# Quantifying information stored in synaptic connections rather than in firing activities of neural networks

**Xinhao Fan**[1*]    **Shreesh P Mysore**[1,2,3*]
[1]Department of Neuroscience
[2]Department of Psychological and Brain Sciences
[3]Kavli Neuroscience Discovery Institute
*Corresponding author
Johns Hopkins University
Baltimore, MD, 21218
{xfan20, mysore}@jhu.edu

## Abstract

A cornerstone of our understanding of both biological and artificial neural networks is that they store information in the strengths of synaptic connections among the constituent neurons. However, in contrast to the well-established theory for quantifying information encoded by the firing activities of neural networks, there does not exist a framework for quantifying information stored by a network's synaptic connections. Here, we develop a theoretical framework using continuous Hopfield networks as an exemplar for associative neural networks, and by modeling real-world data patterns as sets of independent multivariate log-normal distributions. Specifically, we derive, analytically, the Shannon mutual information between the data and singletons, pairs and arbitrary $n$-tuples of synaptic connections within the network. Our framework corroborates well-established insights regarding pattern storage capacity and the principle of distributed coding in neural firing activities. Notably, it discovers synergistic interactions among synapses, revealing that the information encoded jointly by all the synapses exceeds the 'sum of its parts'. Taken together, this study introduces a powerful, interpretable framework for quantitatively understanding information storage in the synapses of neural networks, one that illustrates the duality of synaptic connectivity and neural population activity in learning and memory.

## 1 Introduction

The study of neural networks through the lens of information theory has a long and rich history [1, 2, 3, 4]. Analyses have typically focused on measuring information encoded by the firing activity of neurons in both biological [5, 6] and artificial networks [7, 8]. In contrast, the theoretical analysis of information stored in synaptic connections remains largely underexplored. Some studies analyze upper bounds on "information in weights" by introducing noise into weights [9, 10], but they assume that individual connections are probabilistically independent, neglecting the inherently collective nature of synaptic coding. Alternatively, some research estimates "information per synapse" by dividing capacity by the total number of connections in associative networks such as Hopfield networks [11, 12, 13]. However, this approach relies on firing patterns to infer synaptic coding and, by dividing by synapse count, oversimplifies by assuming uniform efficacy across synapses, failing to account for the diverse roles connections may play.

A major hurdle in quantifying information stored in synaptic weights is the complex and opaque relationship between data and weight distributions in neural networks. Unlike the relatively explicit

mapping between data and neural activities, weight distributions arise implicitly from the network's learning process rather than via a closed-form transformation of data. Consequently, a theoretical analysis of information encoded in ensembles of *connections*, as opposed to ensembles of *cells*, remains largely an open question.

In this study, we propose a foundational framework for synaptic coding based on the mutual information between synaptic connections in a continuous Hopfield network and data patterns assumed to follow a mixture of log-normal distributions (without loss of generality). The accessibility of the weight values in Hopfield networks, and the tractability of the chosen distributions, allow us to derive analytical expressions for the information encoded by individual synapses as well as ensembles ranging from pairs and triplets to larger $n$-tuples. These derivations incorporate two approximations related to the log-normal distribution. The resulting analytical solutions validate established insights about distributed coding and storage capacity from the traditional perspective of network firing activities, and further reveal new insights regarding information synergy among synaptic connections in neural networks. Overall, this research fills an important gap in the field by introducing a theoretical framework for characterizing information storage in neural networks, and highlighting the dual significance of synaptic connectivity and neural population activity in neural coding.

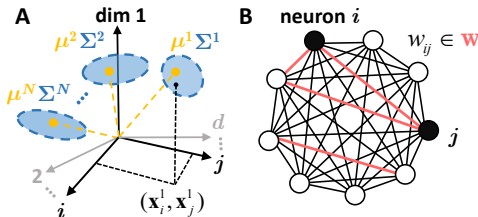

Figure 1: Model setup. (A) Real-world distribution modelled as several independent patterns following multivariate log-normal distribution. (B) Continuous Hopfield network with an example synaptic ensemble $\mathbf{w}$ (red) composed of four connections.

## 2 Analytical framework: Information stored in synaptic connections

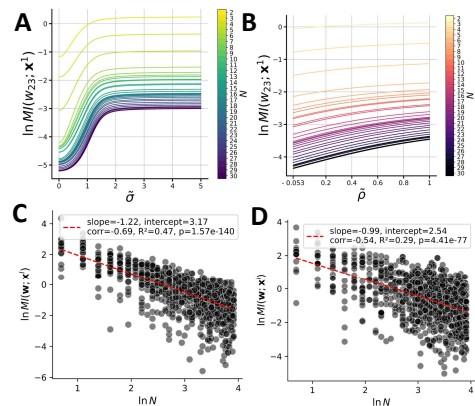

Figure 2: Encoded information as a function of $\tilde{\sigma}$, $\tilde{\rho}$, and the number of data patterns. (A) Information encoded about a specific example pattern ($\mathbf{x}^1$) by a single example weight ($w_{23}$), plotted as a function of $\tilde{\sigma}$ (x-axis) and the number of data patterns $N$ (different colors). (B) Information encoded about $\mathbf{x}^1$ by $w_{23}$, shown as a function of $\tilde{\rho}$ (x-axis) and $N$ (colors). (C, D) This decreasing trend with increasing $N$ also persists for synaptic ensembles $\mathbf{w}$ with randomly sampled configurations (C) shows results for data patterns with the same covariance matrix across samples, while (D) shows results for data patterns with different covariance matrices.

We begin by considering $N$ real-world patterns as independent, $d$-dimensional random variables, denoted as $\{\mathbf{x}^1, \ldots, \mathbf{x}^N\}$. Each $\mathbf{x}^k$ is assumed to follow a multivariate log-normal distribution, formally expressed as: $\ln(\mathbf{x}^k) \sim \mathcal{N}(\mu^k, \Sigma^k)$, $\mathbf{x}^k \perp \mathbf{x}^l$, $\forall k \neq l$, where $\mu^k$ is a $d$-dimensional mean vector in the logarithmic domain, and $\Sigma^k$ is the corresponding $d \times d$ covariance matrix.

For our neural network model, we use the continuous Hopfield network [14], with synaptic weights learned via the Hebbian rule. Specifically, the synaptic connection between neuron $i$ and neuron $j$ is defined as $w_{ij} = \sum_{k=1}^{N} \mathbf{x}_i^k \mathbf{x}_j^k$.

Since $\mathbf{x}_i^k$ and $\mathbf{x}_j^k$ are components of the same multivariate log-normal variable, their product $\mathbf{x}_i^k \mathbf{x}_j^k$ is itself log-normal. Therefore, $w_{ij}$ is a sum of $N$ independent log-normals, which is well-approximated by the Fenton-Wilkinson approach [15]: $\ln w_{ij} \overset{approx.}{\sim} \mathcal{N}(\mu_{w_{ij}}, (\sigma_{w_{ij}})^2)$. This logic extends to any ensemble of $n$ connections $\mathbf{w}$ in the network: since the logarithm of each component is approximately normal, the joint log-vector is approximated as multivariate normal, $\ln \mathbf{w} \overset{approx.}{\sim} \mathcal{N}(\mu_{\mathbf{w}}, \Sigma_{\mathbf{w}})$. With these approximations, we derive the mutual information between the ensemble and an arbitrary data

pattern (pattern $l$) as:

$$MI(\mathbf{w}; \mathbf{x}^l) = \sum_{k=1}^{n} \left( \mu_{w_{ij(k)}} - \mu_{w_{ij(k)/l}} \right) + \frac{1}{2} \ln |\Sigma_{\mathbf{w}}| - \frac{1}{2} \ln |\Sigma_{\mathbf{w}/l}| \tag{1}$$

where $|\Sigma_{\mathbf{w}}|$ denotes the determinant. The parameters $\mu_{w_{ij(k)}}, \mu_{w_{ij(k)/l}}, \Sigma_{\mathbf{w}}, \Sigma_{\mathbf{w}/l}$ are fully specified by the data pattern distributions. Specifically, the set of parameters $\mathcal{S} = \{\mu_i^k, \sigma_{ij}^k \mid k \in \{1, \ldots, N\}, \forall (i, j)$ such that $w_{ij}$ is a component of $\mathbf{w}\}$. For detailed expressions, see Appendix A.

## 3 Results

We performed three simulation sets to analyze how varying the parameter set $\mathcal{S}$ affects $MI$. The first set systematically varies parameters under constraints, while the second and third sets use random sampling and regression analysis to explore broader configurations. Key differences are as follows:

**First Simulation Set:** Parameters were systematically varied with the covariance matrix $\Sigma^k$ fixed and exchangeable across patterns (diagonals: $\tilde{\sigma}^2$; off-diagonals: $\tilde{\rho}\tilde{\sigma}^2$). Results: Figs. 2A, 2B, 3A, 3B.

**Second Simulation Set:** With $\Sigma^k$ still fixed across patterns, a random component ($\mathbf{A}^T\mathbf{A}$, $\mathbf{A}$ random) was added to create more general covariances. Parameters were randomly sampled and analyzed via regression. Results: Figs. 2C, 3C.

**Third Simulation Set:** Here, $\Sigma^k$ varied between patterns, generated from random eigenvalues and orthogonal matrices to remove bias. Again, parameters were randomly sampled with regression analysis. Results: Figs. 2D, 3D, 4.

We also note that while mutual information is reported in nats in theoretical derivations, it is expressed in bits in subsequent analyses for greater interpretability.

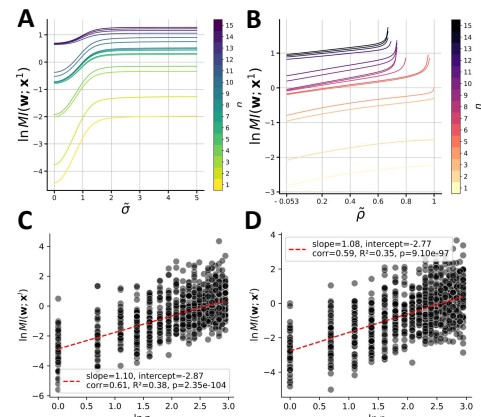

Figure 3: Encoded information and the number of synaptic connections in an ensemble. (A) Mutual information between a sampled ensemble $\mathbf{w}$ and pattern $\mathbf{x}^1$ as a function of $\tilde{\sigma}$ for different ensemble sizes $n$ (colors). (B) Same as (A), but plotted against $\tilde{\rho}$. (C, D) The increase in encoded information with ensemble size persists with random sampling of both $\mathbf{w}$ and other parameters: (C) for data patterns sharing the same covariance matrix, and (D) for data patterns with varying covariance matrices.

### 3.1 Distributed coding in synaptic weights

We find that as the number of stored data patterns increases from 2 to 30, the information carried by each synaptic connection decreases markedly (Fig. 2AB; note the downward shift of curves with increasing pattern number, shown on a log scale). Thus, as more patterns are stored, each connection retains less information about any individual pattern.

This trend holds consistently for synaptic ensembles (Fig. 2C,D). Specifically, mutual information between synaptic weights and patterns, $MI(\mathbf{w}; \mathbf{x}^l)$, shows strong negative correlations with the number of stored patterns—in Fig. 2C, correlation coefficient $r = -0.69$ ($p = 1.57 \times 10^{-140}, R^2 = 0.47$), and in Fig. 2D, $r = -0.54$ ($p = 4.41 \times 10^{-77}, R^2 = 0.29$). Slope analysis reveals that $MI(\mathbf{w}; \mathbf{x}^l)$ scales approximately as $N^{-1.2}$ and $N^{-1.0}$, respectively.

These results indicate that synaptic capacity is distributed among all stored patterns: adding new patterns diminishes the information each connection can hold about previous patterns. Rather than each synapse specializing in a unique pattern—as suggested by the "grandmother connections" hypothesis—connections collectively encode fragments of all patterns. This supports the classic distributed coding paradigm, where information is represented across populations of neurons [16, 17, 18], and extends it to synaptic ensembles.

## 3.2 Capacity analysis for the Hopfield Network

We find that as the number of connections in a synaptic ensemble increases from 1 to 15, the mutual information $MI(\mathbf{w}; \mathbf{x}^1)$ also rises (Fig. 3AB; upward shift from light to dark colors on the log scale) [1]. This pattern persists across different ensemble configurations (Fig. 3C,D): $MI(\mathbf{w}; \mathbf{x}^l)$ shows strong positive correlations with ensemble size ($r = 0.61$, $p = 2.35 \times 10^{-104}$ in Fig. C; $r = 0.59$, $p = 9.10 \times 10^{-97}$ in Fig. D), and scaling follows $MI \sim n^{1.1}$, where $n$ is the number of connections.

With all synaptic connections included ($n \sim d^2$ for $d$ neurons in a Hopfield network), the total stored information scales as $MI \sim d^{2.2}$. Because the information needed to encode one pattern is $H_{\text{pattern}} \sim d$, the number of patterns that can be stored is estimated by $P \sim MI/H_{\text{pattern}} \sim d^{1.2}$.

Unlike classical results for discrete Hopfield networks, which typically yield $P \sim d$ or below [19, 20], our analysis suggests a superlinear scaling for continuous networks. This difference arises because our approach quantifies information directly in connectivity, while traditional assessments evaluate capacity based on pattern retrieval through firing dynamics. According to the data-processing inequality, information encoded in the weights exceeds that retrievable by network dynamics, so our results likely reflect the maximum possible storage before retrieval loss.

## 3.3 Synergy among synaptic connections

To examine how the contribution of a single connection varies with ensemble size, we analyzed $MI(\mathbf{w}, \mathbf{x}^l)/n$. This metric reveals how individual synapses behave within larger groups. Figure 4A shows that the distribution of $\ln(MI(\mathbf{w}, \mathbf{x}^l)/n)$ becomes increasingly right-shifted as $n$ increases from 1 to 4 to 16. The differences between $n = 1$ and $n = 4$ ($p = 5.7 \times 10^{-5}$) and between $n = 4$ and $n = 16$ ($p = 8.5 \times 10^{-5}$) are statistically significant. Regression analysis (Fig. 4B) also reveals a strong positive correlation between ensemble size $n$ and the mean logarithm of per-connection information ($r = 0.70$, $p = 8.55 \times 10^{-4}$, $R^2 = 0.49$), indicating that each connection encodes more information as part of a larger ensemble. In other words, $\mathbb{E}[MI(\mathbf{w}; \mathbf{x}^l)]/n > \mathbb{E}[MI(w; \mathbf{x}^l)]$ for $n > 1$.

These findings demonstrate that the information collectively stored by synaptic ensembles exceeds the sum of information stored by individual connections alone, i.e., $\mathbb{E}[MI(\mathbf{w}; \mathbf{x}^l)] >$

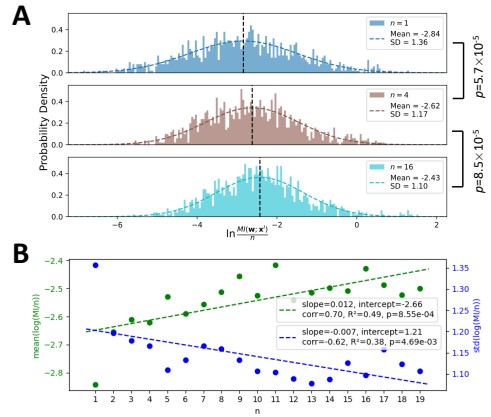

Figure 4: Information contributed per synaptic connection as a function of ensemble size. (A) Histograms showing the distribution of $\ln(MI/n)$ for $n = 1, 4, 9, 16$. (B) The relationship between $n$ and the mean (green, left axis) as well as standard deviation (blue, right axis) of $\ln(MI/n)$.

$n \, \mathbb{E}[MI(w; \mathbf{x}^l)]$. This phenomenon, known as *information synergy* [21, 22, 23], reflects that connections working together can encode more than the sum of their parts. While synergy has long been recognized among groups of neurons or brain regions [24, 25, 26], our approach reveals that this principle extends to the level of synaptic connections themselves.

# 4 Discussion

## 4.1 Choice of continuous Hopfield network

In this initial exploration of information encoding in synaptic ensembles, we employ the continuous Hopfield network with Hebbian-like connectivity for its mathematical tractability. We do not address the added complexity present in more general architectures of artificial or biological networks.

---

[1]In Fig. 3B, for ensembles with five or more connections, large $\tilde{\rho}$ values can produce NaN or infinite results; affected curves are truncated at this point. Similar issues in regression were addressed by extrapolation (see Materials and Methods, Section B.2 for details).

Nonetheless, the Hopfield network is a fundamental model, widely used to represent dynamics in various brain regions [27, 28, 29, 30, 31]. Thus, our framework marks a meaningful step towards characterizing information storage in synaptic connections.

## 4.2 Assumptions on real world data distribution

The assumption that data follow log-normal distributions may seem restrictive, but is justified since Shannon mutual information is invariant under smooth, bijective transformations such as translation and scaling [32]. As long as the true data distribution can be smoothly transformed into a log-normal form, our analysis remains applicable. Therefore, our results are likely robust for any unimodal, skewed distribution with long tails—a property common in many natural data types, including images [33], sounds [34], and language [35]. We believe these characteristics indicate that the insights from our framework extend beyond our specific modeling assumptions and are broadly applicable to synaptic information encoding.

## 4.3 Decoding information from network dynamics

In this work, we focused on quantifying the information encoded in the network weights, without considering the retrieval dynamics required to decode that information. Consequently, our capacity analysis should be interpreted as an upper bound. Nevertheless, even if certain information cannot be recovered through the classical retrieval dynamics of a Hopfield network, other mechanisms may still enable full information retrieval.

Specifically, decoding could be accomplished by specialized downstream neural circuits. For example, one might probe the learned network by injecting external signals and then analyze the resulting dynamics to infer the stored information. To illustrate this idea, consider a continuous Hopfield network characterized by a connectivity matrix $W$, activation function $f$, and time constant $\tau$. When a signal $\mathbf{I}(t)$ is applied, the network's firing dynamics follow:

$$\tau \dot{\mathbf{x}}(t) \; = \; -\mathbf{x}(t) \; + \; f\big(W\mathbf{x}(t) + \mathbf{I}(t)\big) \tag{2}$$

This expression can be reformulated in a linear regression form as:

$$\mathbf{y}(t) := f^{-1}\big(\tau \dot{\mathbf{x}}(t) + \mathbf{x}(t)\big) - \mathbf{I}(t), \tag{3}$$

$$\mathbf{y}(t) \; = W\mathbf{x}(t) \tag{4}$$

By observing $Y := [\mathbf{y}(1) \; \cdots \; \mathbf{y}(t)]$ and $X := [\mathbf{x}(1) \; \cdots \; \mathbf{x}(t)]$, we obtain: $W = YX^\top(XX^\top)^{-1}$. The invertibility of $XX^\top$ can be ensured by appropriately controlling $\mathbf{I}(t)$ such that $\mathbf{x}(t)$ spans the entire state space. This observation suggests that, in principle, it is possible to recover the full connectivity matrix $W$ from the firing dynamics. However, this represents only one hypothetical scenario. Determining how the brain actually decodes the information embedded in synaptic weights remains an open question for future research.

## 4.4 Implications for modern deep learning

In modern deep learning, networks often employ multilayered architectures and non-Hebbian learning rules. Nevertheless, our framework may still provide useful insights. If we focus on any two adjacent layers in a deep neural network, these two layers—together with the weights connecting them—resemble a continuous Hopfield network, except that the within-layer connectivity is absent.

Let the activities of the two layers (for simplicity, referred to as the input and output layers) be denoted by $\mathbf{x}$ and $\mathbf{y}$, respectively. Depending on the learning setup $f$, the connections can in some cases be expressed as $w_{ij} = \sum_t f(\mathbf{x}_j, \mathbf{y}_i, t)$, where the summation is taken over training steps $t$. For Hebbian learning, this leads to $w_{ij} = \sum_t \mathbf{x}_j \mathbf{y}_i$.

Following the same reasoning as in our analysis of Hopfield networks, we can quantify how much information the weight ensemble encodes about the co-activity patterns of the input and output layers. Specifically, $MI(\mathbf{w}; \mathbf{x}, \mathbf{y}) = MI(\mathbf{w}; \mathbf{x}) + MI(\mathbf{w}; \mathbf{y}|\mathbf{x})$, which decomposes the total information in the weights into two parts: information about the input activity $\mathbf{x}$, and information that helps predict the output $\mathbf{y}$ given the input.

We therefore speculate that the phenomena revealed by our framework—such as distributed coding and synergistic information—may also emerge between adjacent layers in modern deep neural networks.

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

# A Appendix: Model setup

## A.1 Information encoded by one synaptic connection

Here, we derive the analytical expression for the mutual information $MI(w_{ij}; \mathbf{x}^l)$ between a synaptic connection $w_{ij}$ and a data pattern $\mathbf{x}^l$, by obtaining closed-form expressions for the two pertinent probability distributions: $p(w_{ij})$ and $p(w_{ij}|\mathbf{x}^l)$.

For a given pattern $\mathbf{x}^k$, the vector $\ln \mathbf{x}^k$ follows a multivariate Gaussian distribution. Consequently, its $i^{\text{th}}$ component also follows a Gaussian distribution, denoted as $\ln \mathbf{x}_i^k \sim \mathcal{N}(\mu_i^k, (\sigma_i^k)^2)$, where the mean $\mu_i^k$ is the $i^{\text{th}}$ component of $\mu^k$, and the variance $(\sigma_i^k)^2$ is the $i^{\text{th}}$ diagonal element of the covariance matrix $\Sigma^k$. Similarly, the sum of the $i^{\text{th}}$ and $j^{\text{th}}$ components follows a Gaussian distribution of the form:

$$\ln(\mathbf{x}_i^k) + \ln(\mathbf{x}_j^k) = \ln(\mathbf{x}_i^k \mathbf{x}_j^k) \sim \mathcal{N}(\mu_i^k + \mu_j^k, (\sigma_i^k)^2 + (\sigma_j^k)^2 + 2\sigma_{ij}^k), \tag{5}$$

where $\sigma_{ij}^k$ represents the $(i, j)$ element of the covariance matrix $\Sigma^k$. Therefore, the random variable $\mathbf{x}_i^k \mathbf{x}_j^k$ also follows a log-normal distribution. The synaptic connection $w_{ij} = \sum_{k=1}^N \mathbf{x}_i^k \mathbf{x}_j^k$ is thus the sum of multiple independent log-normally distributed variables.

Fortunately, the problem of approximating the sum of log-normal variables has been studied extensively in engineering due to the prevalence of such noise in communication systems. A widely used approach in the field, Fenton-Wilkinson approximation [15], demonstrates that the sum of log-normal variables can be reasonably well approximated by another log-normal variable using the moment-matching method. We apply this approach to derive the probability distribution $p(w_{ij})$.

**Proposition 1.1.** *For $N$ independent log-normally distributed patterns $\{\mathbf{x}^1, ..., \mathbf{x}^N\}$ with $\ln \mathbf{x}_i^k \sim \mathcal{N}(\mu_i^k, (\sigma_i^k)^2)$, the probability distribution of the synaptic connection $w_{ij} := \sum_{k=1}^N \mathbf{x}_i^k \mathbf{x}_j^k$ in the Hopfield network can be approximated using a new log-normal distribution denoted by:*

$$\ln w_{ij} \overset{approx.}{\sim} \mathcal{N}(\mu_{w_{ij}}, (\sigma_{w_{ij}})^2) \tag{6}$$

*i.e.,*

$$p(w_{ij}) \approx \frac{1}{w_{ij}\sigma_{w_{ij}}\sqrt{2\pi}} \exp\left(-\frac{(\ln w_{ij} - \mu_{w_{ij}})^2}{2\sigma_{w_{ij}}^2}\right) \tag{7}$$

*with the parameters $\mu_{w_{ij}}$ and $\sigma_{w_{ij}}^2$ being:*

$$\mu_{w_{ij}} = \ln M_1 - \frac{1}{2}\ln(1 + \frac{M_2}{M_1^2}), \quad \sigma_{w_{ij}}^2 = \ln(1 + \frac{M_2}{M_1^2}) \tag{8}$$

*and with $M_1$ and $M_2$ defined as:*

$$M_1 := \sum_{k=1}^N \exp\left(\mu_i^k + \mu_j^k + \frac{1}{2}\left[(\sigma_i^k)^2 + (\sigma_j^k)^2 + 2\sigma_{ij}^k\right]\right) \tag{9}$$

$$M_2 := \sum_{k=1}^N \left[\exp\left((\sigma_i^k)^2 + (\sigma_j^k)^2 + 2\sigma_{ij}^k\right) - 1\right]\exp\left(2(\mu_i^k + \mu_j^k) + (\sigma_i^k)^2 + (\sigma_j^k)^2 + 2\sigma_{ij}^k\right) \tag{10}$$

Next, we derive the expression for the conditional distribution $p(w_{ij}|\mathbf{x}^l)$. Given the independence of the $N$ terms defining a given synaptic weight, i.e., $\mathbf{x}_i^k \mathbf{x}_j^k \perp \mathbf{x}_i^l \mathbf{x}_j^l \; \forall k \neq l$, the distributions $p(w_{ij}|\mathbf{x}^l)$ and $p\left(\sum_{k\neq l}^N \mathbf{x}_i^k \mathbf{x}_j^k\right)$ are isomorphic in the sense that the latter is a shifted version of the former by a constant $-\mathbf{x}_i^l \mathbf{x}_j^l$. For convience, we denote $w_{ij/l} := \sum_{k\neq l}^N \mathbf{x}_i^k \mathbf{x}_j^k$. Consequently, we can derive the probability distribution for $w_{ij/l}$ instead, and do so using the same derivation steps as for $w_{ij}$. We denote this distribution by:

$$\ln w_{ij/l} := \ln\left(\sum_{k\neq l}^N \mathbf{x}_i^k \mathbf{x}_j^k\right) \overset{\text{approx.}}{\sim} \mathcal{N}\left(\mu_{w_{ij/l}}, (\sigma_{w_{ij/l}})^2\right), \tag{11}$$

where the parameters $\mu_{w_{ij/l}}$ and $(\sigma_{w_{ij/l}})^2$ have the same form as in Equation 8, with the $l^{\text{th}}$ term removed from the summation in Equations 9 and 10. For detailed expressions, see section A.4.

With both $p(w_{ij})$ and $p(w_{ij}|\mathbf{x}^l)$ known, we can obtain the analytical expression for the mutual information.

**Proposition 1.2.** *The mutual information between a synaptic connection and a data pattern is given by:*

$$MI(w_{ij};\mathbf{x}^l) = \left(\mu_{w_{ij}} - \mu_{w_{ij/l}}\right) + \left(\ln \sigma_{w_{ij}} - \ln \sigma_{w_{ij/l}}\right), \tag{12}$$

*with the units in nats.*

We note, here, that the mutual information between a synaptic connection $w_{ij}$ and a data pattern $\mathbf{x}^l$ (eqn. 12), depends on the distribution properties of each data pattern, specifically their means and covariance matrices (eqns. 8, 9, 10), as well as the specific connection weight $w_{ij}$ (eqn. 7) and the choice of pattern $\mathbf{x}^l$.

## A.2   Information encoded by a pair of connections

Synaptic connections within a neural network may impact one another indirectly, thereby influencing coding beyond the information contained in a single connection. To study the collective coding by these connections, we focus first on the mutual information between data and the joint activity of two connections, $MI(\mathbf{w};\mathbf{x}^l)$, where $\mathbf{w} = (w_{ij}, w_{mn})^T$. This analysis requires knowledge of the distributions $p(\mathbf{w})$ and $p(\mathbf{w}|\mathbf{x}^l)$.

From the derivation in the single connection case, we obtain the marginals for each weight, which are given by:

$$\ln w_{ij} \overset{\text{approx.}}{\sim} \mathcal{N}\left(\mu_{w_{ij}},(\sigma_{w_{ij}})^2\right), \quad \ln w_{mn} \overset{\text{approx.}}{\sim} \mathcal{N}\left(\mu_{w_{mn}},(\sigma_{w_{mn}})^2\right), \tag{13}$$

This result indicates that the individual components of the vector $\ln \mathbf{w} = (\ln w_{ij}, \ln w_{mn})^T$ are marginally Gaussian. While this does not necessarily imply that the joint distribution is multivariate Gaussian, we follow a common approximation for tractability by modeling the joint distribution as bivariate Gaussian here. Specifically, we express the joint distribution of $\ln \mathbf{w}$ as:

$$\ln \mathbf{w} = \begin{pmatrix} \ln w_{ij} \\ \ln w_{mn} \end{pmatrix} \overset{\text{approx.}}{\sim} \mathcal{N}(\mu_{\mathbf{w}} = \begin{pmatrix} \mu_{w_{ij}} \\ \mu_{w_{mn}} \end{pmatrix}, \Sigma_{\mathbf{w}} = \begin{pmatrix} \sigma^2_{w_{ij}} & \sigma_{w_{ij,mn}} \\ \sigma_{w_{mn,ij}} & \sigma^2_{w_{mn}} \end{pmatrix}). \tag{14}$$

The only remaining unknown in this equation is $\sigma_{w_{ij,mn}}$, which represents the covariance between $\ln w_{ij}$ and $\ln w_{mn}$. The expression for this covariance is presented in the following proposition:

**Proposition 2.1.** *Under the approximation that a pair of synaptic connections $(w_{ij}, w_{mn})^T := (\sum_k^N \mathbf{x}_i^k\mathbf{x}_j^k, \sum_k^N \mathbf{x}_m^k\mathbf{x}_n^k)^T$ follows a 2-dimensional log-normal distribution, the covariance between $\ln w_{ij}$ and $\ln w_{mn}$ is:*

$$\sigma_{w_{ij,mn}} = \ln(1 +$$

$$\frac{\sum_{k=1}^N \exp\left(\sum_\phi^{\{i,j,m,n\}} \mu_\phi^k + \frac{1}{2}\sum_\phi^{\{i,j,m,n\}}(\sigma_\phi^k)^2 + \sum_\phi^{\{ij,mn\}} \sigma_\phi^k\right)\left[\exp\left(\sum_\phi^{\{i,j\}\times\{m,n\}} \sigma_\phi^k\right) - 1\right]}{\sum_{k,l=1}^N \exp\left(\sum_\phi^{\{i,j\}} \mu_\phi^k + \sum_\phi^{\{m,n\}} \mu_\phi^l + \frac{1}{2}\sum_\phi^{\{i,j\}}(\sigma_\phi^k)^2 + \frac{1}{2}\sum_\phi^{\{m,n\}}(\sigma_\phi^l)^2 + \sigma_{ij}^k + \sigma_{mn}^l\right)})$$

$$\tag{15}$$

*with the summation notation meaning:*

$$\sum_\phi^{\{ij,mn\}} \sigma_\phi^k = \sigma_{ij}^k + \sigma_{mn}^k, \quad \sum_\phi^{\{i,j\}\times\{m,n\}} \sigma_\phi^k = \sigma_{im}^k + \sigma_{in}^k + \sigma_{jm}^k + \sigma_{jn}^k \tag{16}$$

These derivations show how the distribution $p(\mathbf{w})$ is approximated by a 2-dimensional log-normal distribution with derived parameter values. The next step is to study $p(\mathbf{w}|\mathbf{x}^l)$. Similar to the single connection scenario, $p(\mathbf{w}|\mathbf{x}^l)$ and $p\left(\sum_{k\neq l}^N \mathbf{x}_i^k\mathbf{x}_j^k, \sum_{k\neq l}^N \mathbf{x}_m^k\mathbf{x}_n^k\right)$ are isomorphic distributions, with their domains shifted. For convenience, we denote $\mathbf{w}_{/l} := (\sum_{k\neq l}^N \mathbf{x}_i^k\mathbf{x}_j^k, \sum_{k\neq l}^N \mathbf{x}_m^k\mathbf{x}_n^k)^T$ for two

connections case. Therefore, the same steps used to derive $p(\mathbf{w})$ apply here, leading to the following distribution:

$$\ln \mathbf{w}_{/l} := \begin{pmatrix} \ln w_{ij/l} \\ \ln w_{mn/l} \end{pmatrix} \overset{\text{approx.}}{\sim} \mathcal{N}(\mu_{\mathbf{w}_{/l}} = \begin{pmatrix} \mu_{w_{ij/l}} \\ \mu_{w_{mn/l}} \end{pmatrix}, \Sigma_{\mathbf{w}_{/l}} = \begin{pmatrix} \sigma^2_{w_{ij/l}} & \sigma_{w_{ij,mn/l}} \\ \sigma_{w_{mn,ij/l}} & \sigma^2_{w_{mn/l}} \end{pmatrix}), \quad (17)$$

where the parameters with $/l$ are equal to their original corresponding parameters, with the terms related to $\mathbf{x}^l$ removed from the summation. For a more detailed expression for $\mu_{\mathbf{w}_{/l}}$ and $\Sigma_{\mathbf{w}_{/l}}$, please refer to the Methods section A.4.

Given $p(\mathbf{w})$ and $p(\mathbf{w}|\mathbf{x}^l)$, we can then calculate the mutual information encoded by any pair of synaptic connections about a data pattern, as stated in the following proposition:

**Proposition 2.2.** *The mutual information between a pair of connections $\mathbf{w} = (w_{ij}, w_{mn})^T$ and a data pattern $\mathbf{x}^l$ is given by:*

$$MI(\mathbf{w}; \mathbf{x}^l) = (\mu_{w_{ij}} + \mu_{w_{mn}} - \mu_{w_{ij/l}} - \mu_{w_{mn/l}}) + \frac{1}{2} \ln |\Sigma_{\mathbf{w}}| - \frac{1}{2} \ln |\Sigma_{\mathbf{w}_{/l}}|, \quad (18)$$

*where $|\Sigma_{\mathbf{w}}|$ denotes the determinant of the matrix $\Sigma_{\mathbf{w}}$. The information is expressed in nats.*

### A.3 Information in an ensemble of multiple synaptic connections

The next step is to extend the analysis to the case of information encoded jointly in an ensemble of $n$ synaptic connections, denoted as $\mathbf{w} = (w_{ij(1)}, w_{ij(2)}, \ldots, w_{ij(n)})^T$. Here, the set $\{ij(1), ij(2), \ldots, ij(n)\}$ represents the indices for the $n$ synaptic weights. For example, $w_{ij(k)}$ refers to the connection between neuron $i(k)$ and neuron $j(k)$. The mutual information encoded by any $n$ connections about the data, $MI(\mathbf{w}; \mathbf{x}^l)$, can be derived using a similar approach as before. Since all marginals for $\ln \mathbf{w}$ are normally distributed, we approximate the distribution as an $n$-dimensional log-normal distribution, leading to:

$$\ln \mathbf{w} = \begin{pmatrix} \ln w_{ij(1)} \\ \vdots \\ \ln w_{ij(n)} \end{pmatrix} \overset{\text{approx.}}{\sim} \mathcal{N}(\mu_{\mathbf{w}} = \begin{pmatrix} \mu_{w_{ij(1)}} \\ \vdots \\ \mu_{w_{ij(n)}} \end{pmatrix}, \Sigma_{\mathbf{w}} = \begin{pmatrix} \sigma^2_{w_{ij(1)}} & \sigma_{w_{ij(1),ij(2)}} & \cdots & \sigma_{w_{ij(1),ij(n)}} \\ \sigma_{w_{ij(2),ij(1)}} & \sigma^2_{w_{ij(2)}} & \cdots & \sigma_{w_{ij(2),ij(n)}} \\ \vdots & \vdots & \ddots & \vdots \\ \sigma_{w_{ij(n),ij(1)}} & \sigma_{w_{ij(n),ij(2)}} & \cdots & \sigma^2_{w_{ij(n)}} \end{pmatrix})$$
$$(19)$$

This expression represents $p(\mathbf{w})$ as an $n$-dimensional log-normal distribution. The parameters $\mu_{\mathbf{w}}$ and the diagonal elements of $\Sigma_{\mathbf{w}}$ follow the form given in Equation 8, while the off-diagonal elements of $\Sigma_{\mathbf{w}}$ follow Equation 15, specifically:

$$\sigma_{w_{ij(r),ij(s)}} = \ln(1 +$$
$$\frac{\sum_{k=1}^{N} \exp\left(\sum_m^{\{r,s\}} \left[\sum_\phi^{\{i(m),j(m)\}} \mu_\phi^k + \frac{1}{2}(\sigma_\phi^k)^2\right] + \sigma_{ij(m)}^k\right) \left[\exp\left(\sum_\phi^{\{i(r),j(r)\} \times \{i(s),j(s)\}} \sigma_\phi^k\right) - 1\right]}{\sum_{k,l=1}^{N} \exp\left(\left[\sum_\phi^{\{i(r),j(r)\}} \mu_\phi^k + \frac{1}{2}(\sigma_\phi^k)^2\right] + \left[\sum_\phi^{\{i(s),j(s)\}} \mu_\phi^l + \frac{1}{2}(\sigma_\phi^l)^2\right] + \sigma_{ij(r)}^k + \sigma_{ij(s)}^l\right)})$$
$$(20)$$

To obtain $p(\mathbf{w}|\mathbf{x}^l)$, as in the single and double connection scenarios, we focus on its isomorphic distribution $p(\mathbf{w}_{/l})$ with $\mathbf{w} = (w_{ij(1)/l}, w_{ij(2)/l}, \ldots, w_{ij(n)/l})^T$, which is also treated as log-normally distributed. The parameters with $/l$ are simply the original parameters with the summation terms related to $\mathbf{x}^l$ excluded:

$$\ln \mathbf{w}_{/l} = \begin{pmatrix} \ln w_{ij(1)/l} \\ \vdots \\ \ln w_{ij(n)/l} \end{pmatrix} \overset{\text{approx.}}{\sim} \mathcal{N}(\mu_{\mathbf{w}_{/l}} = \begin{pmatrix} \mu_{w_{ij(1)/l}} \\ \vdots \\ \mu_{w_{ij(n)/l}} \end{pmatrix}, \Sigma_{\mathbf{w}_{/l}} = \begin{pmatrix} \sigma^2_{w_{ij(1)/l}} & \cdots & \sigma_{w_{ij(1),ij(n)/l}} \\ \vdots & \ddots & \vdots \\ \sigma_{w_{ij(n),ij(1)/l}} & \cdots & \sigma^2_{w_{ij(n)/l}} \end{pmatrix})$$
$$(21)$$

Please refer to section A.4 for detailed expressions for $\mu_{\mathbf{w}_{/l}}$ and $\Sigma_{\mathbf{w}_{/l}}$.

Finally, with $p(\mathbf{w})$ and $p(\mathbf{w}|\mathbf{x}^l)$, we can derive the analytical expression for the information jointly encoded in multiple synaptic connections about a pattern:

**Proposition 3.** *The mutual information between the joint activity of $n$ synaptic connections $\mathbf{w} = (w_{ij(1)}, w_{ij(2)}, ..., w_{ij(n)})^T$ and a data pattern $\mathbf{x}^l$ is:*

$$MI(\mathbf{w}; \mathbf{x}^l) = \sum_{k=1}^{n} \left( \mu_{w_{ij(k)}} - \mu_{w_{ij(k)/l}} \right) + \frac{1}{2} \ln |\Sigma_{\mathbf{w}}| - \frac{1}{2} \ln |\Sigma_{\mathbf{w}/l}| \tag{22}$$

After all the derivations, we obtain an explicit expression showing how the information encoded in a specified synaptic ensemble about a particular data pattern, $MI(\mathbf{w}|\mathbf{x}^l)$, is fully determined by the parameters of the distributions for all $N$ real-world data patterns. Specifically, we observe that $MI(\mathbf{w}|\mathbf{x}^l)$ depends solely on the means and covariances of the relevant components of these data patterns—those components corresponding to the neurons involved in the ensemble. Formally, this dependence can be summarized by the set $\mathcal{S} = \{\mu_i^k, \sigma_{ij}^k \mid k \in \{1, \ldots, N\}, \forall (i,j)$ such that $w_{ij}$ is a component of $\mathbf{w}\}$. Consequently, both the size of the ensemble, $n$, and the particular choice of connections included in the ensemble, influence $MI$, as they determine the elements included in the set $\mathcal{S}$.

Importantly, $MI(\mathbf{w}|\mathbf{x}^l)$ depends only on the parameters associated with the neurons participating in the ensemble; neurons in the network that lie outside the ensemble have no effect. This finding implies that as long as the local statistics, $\mathcal{S}$, of the specified neurons remain fixed, the value of $MI(\mathbf{w}|\mathbf{x}^l)$ will remain unchanged, regardless of the larger Hopfield network in which the ensemble is embedded. In this sense, our result is context-invariant: it depends solely on the properties of the target ensemble and is independent of the total size, $d$, of the network containing the ensemble.

### A.4 Proofs for propositions

**Proposition 1.1.** *For $N$ independent log-normally distributed patterns $\{\mathbf{x}^1, ..., \mathbf{x}^N\}$, the distribution of the synaptic connection $w_{ij} = \sum_{k=1}^{N} \mathbf{x}_i^k \mathbf{x}_j^k$ in the Hopfield network is approximated by a new log-normal distribution:*

$$p(w_{ij}) \approx \frac{1}{w_{ij} \sigma_{w_{ij}} \sqrt{2\pi}} \exp\left( -\frac{(\ln w_{ij} - \mu_{w_{ij}})^2}{2\sigma_{w_{ij}}^2} \right) \tag{23}$$

*with $\mu_{w_{ij}}, \sigma_{w_{ij}}$ defined in equation 28, 29.*

**Proof.** Each data pattern is assumed to follow $\ln(\mathbf{x}^k) \sim \mathcal{N}(\mu^k, \Sigma^k)$ in equation **??**. Since the sum of two components in a multivariate normal distribution is normal, we have:

$$\ln(\mathbf{x}_i^k \mathbf{x}_j^k) = \ln(\mathbf{x}_i^k) + \ln(\mathbf{x}_j^k) \sim \mathcal{N}(\mu_i^k + \mu_j^k, (\sigma_i^k)^2 + (\sigma_j^k)^2 + 2\sigma_{ij}^k), \tag{24}$$

which means any term of the product form $\mathbf{x}_i^k \mathbf{x}_j^k$ follows a log-normal distribution. The sum of such terms from $N$ independent data patterns, $\sum_{k=1}^{N} \mathbf{x}_i^k \mathbf{x}_j^k$, represents the connection weight between two neurons, $w_{ij}$, as defined as in equation **??**. Therefore, studying the distribution of $w_{ij}$ is equivalent to studying the distribution of the sum of several independent log-normal random variables. Using Fenton-Wilkinson method, we can approximate this distribution, $p(w_{ij})$, by a new log-normal distribution $p(\hat{w}_{ij}) \approx p(w_{ij})$ with the same first and second moments. In other words, we aim to derive a log-normal distribution:

$$\ln(\hat{w}_{ij}) \sim \mathcal{N}\left( \mu_{w_{ij}}, (\sigma_{w_{ij}})^2 \right) \tag{25}$$

whose first two moments match those of $w_{ij}$:

$$\mathbb{E}[\hat{w}_{ij}] = \mathbb{E}[w_{ij}], \ \text{Var}[\hat{w}_{ij}] = \text{Var}[w_{ij}].$$

The values for $\mathbb{E}[w_{ij}]$ and $\text{Var}[w_{ij}]$ can be derived based on independence among patterns and the properties of the log-normal distribution[2]. For convenience, we denote them as $M_1$ and $M_2$ from

---

[2]For random variable $\ln X \sim \mathcal{N}(\mu, \sigma^2)$, $\mathbb{E}[X] = e^{\mu + \frac{1}{2}\sigma^2}$, $\text{Var}[X] = [e^{\sigma^2} - 1]e^{2\mu + \sigma^2}$.

here on.

$$M_1 := \mathbb{E}[w_{ij}] = \mathbb{E}[\sum_{k=1}^{N} \mathbf{x}_i^k \mathbf{x}_j^k] = \sum_{k=1}^{N} \mathbb{E}[\mathbf{x}_i^k \mathbf{x}_j^k]$$

$$= \sum_{k=1}^{N} \exp\left( \mu_i^k + \mu_j^k + \frac{1}{2}\left[ (\sigma_i^k)^2 + (\sigma_j^k)^2 + 2\sigma_{ij}^k \right] \right) \tag{26}$$

$$M_2 := \mathrm{Var}[w_{ij}] = \mathrm{Var}[\sum_{k=1}^{N} \mathbf{x}_i^k \mathbf{x}_j^k] = \sum_{k=1}^{N} \mathrm{Var}[\mathbf{x}_i^k \mathbf{x}_j^k]$$

$$= \sum_{k=1}^{N} \left[ \exp\left( (\sigma_i^k)^2 + (\sigma_j^k)^2 + 2\sigma_{ij}^k \right) - 1 \right] \exp\left( 2(\mu_i^k + \mu_j^k) + (\sigma_i^k)^2 + (\sigma_j^k)^2 + 2\sigma_{ij}^k \right) \tag{27}$$

Given $\mathbb{E}[\hat{w}_{ij}]$ and $\mathrm{Var}[\hat{w}_{ij}]$, the the parameters $\mu_{w_{ij}}$ and $\sigma^2_{w_{ij}}$ in equation 25 are derived as follows:

$$\mu_{w_{ij}} = \ln M_1 - \frac{1}{2}\ln(1 + \frac{M_2}{M_1{}^2}), \tag{28}$$

$$\sigma^2_{w_{ij}} = \ln(1 + \frac{M_2}{M_1{}^2}). \tag{29}$$

In this way, we obtain a distribution for a connection weight that is fully parametrized by the known data pattern distributions. $\qquad\square$

**Proposition 1.2.** *The mutual information between a synaptic connection $w_{ij}$ and a data pattern $\mathbf{x}^l$ is given by:*

$$MI(w_{ij}; \mathbf{x}^l) = \left( \mu_{w_{ij}} - \mu_{w_{ij/l}} \right) + \left( \ln \sigma_{w_{ij}} - \ln \sigma_{w_{ij/l}} \right),$$

*with $\mu_{w_{ij}}, \sigma_{w_{ij}}$ defined in equation 28, 29; $\mu_{w_{ij/l}}, \sigma_{w_{ij/l}}$ defined in equation 30, 31. The unit of $MI$ is nat.*

**Proof.** We can calculate the mutual information $MI(w_{ij}; \mathbf{x}^l)$ as the difference of differential entropies $h(w_{ij}) - h(w_{ij}|\mathbf{x}^l)$, which leads to:

$$MI(w_{ij}; \mathbf{x}^l) = h(w_{ij}) - h(w_{ij}|\mathbf{x}^l)$$

$$= h(w_{ij}) + \iint p(\mathbf{x}^l)p(w_{ij}|\mathbf{x}^l) \log p(w_{ij}|\mathbf{x}^l) \, \mathrm{d}w_{ij}\mathrm{d}\mathbf{x}^l$$

$$= h(w_{ij}) + \int p(\mathbf{x}^l) \left( \int p(w_{ij}|\mathbf{x}^l) \log p(w_{ij}|\mathbf{x}^l) \, \mathrm{d}w_{ij} \right) \mathrm{d}\mathbf{x}^l$$

$$= h(w_{ij}) - \int p(\mathbf{x}^l) h(\sum_{k \neq l}^{N} \mathbf{x}_i^k \mathbf{x}_j^k) \mathrm{d}\mathbf{x}^l$$

$$= h(w_{ij}) - h(\sum_{k \neq l}^{N} \mathbf{x}_i^k \mathbf{x}_j^k)$$

In the second-to-last step, we use the fact that $p(w_{ij}|\mathbf{x}^l)$ and $p(\sum_{k \neq l}^{N} \mathbf{x}_i^k \mathbf{x}_j^k)$ have the same differential entropy. This is because, given independence among patterns, $p(w_{ij}|\mathbf{x}^l) := p(\sum_{k}^{N} \mathbf{x}_i^k \mathbf{x}_j^k|\mathbf{x}^l)$ and $p(\sum_{k \neq l}^{N} \mathbf{x}_i^k \mathbf{x}_j^k)$ are shifted versions of each other. For convenience, we denote $w_{ij/l} := \sum_{k \neq l}^{N} \mathbf{x}_i^k \mathbf{x}_j^k$ in what follows. Observe that the same reasoning represented in Proposition 1.1 also extends to $w_{ij/l}$, indicating it too can be also approximated by a log-normal distribution.

The $MI(w_{ij}; \mathbf{x}^l)$ is therefore the difference between the differential entropies of two log-normal variables, $w_{ij}$ and $w_{ij/l}$. Considering the property of log-normal random variables[3], we have:

$$h(w_{ij}) = \ln(\sqrt{2\pi}\sigma_{w_{ij}} e^{\mu_{w_{ij}} + \frac{1}{2}}),$$

$$h(w_{ij/l}) = \ln(\sqrt{2\pi}\sigma_{w_{ij/l}} e^{\mu_{w_{ij/l}} + \frac{1}{2}}).$$

---

[3]For random variable $\ln X \sim \mathcal{N}(\mu, \sigma^2)$, its entropy equals $\ln(\sqrt{2\pi}\sigma e^{\mu + \frac{1}{2}})$ in nats.

Consequently, we obtain the expression for mutual information:

$$MI(w_{ij}; \mathbf{x}^l) = \ln(\sqrt{2\pi}\sigma_{w_{ij}}e^{\mu_{w_{ij}}+\frac{1}{2}}) - \ln(\sqrt{2\pi}\sigma_{w_{ij/l}}e^{\mu_{w_{ij/l}}+\frac{1}{2}})$$

$$= \left(\mu_{w_{ij}} - \mu_{w_{ij/l}}\right) + \left(\ln\sigma_{w_{ij}} - \ln\sigma_{w_{ij/l}}\right)$$

For the sake of completeness, we explicitly write the parameters for $p(w_{ij/l})$ here. By adapting equation 25 to 29, we have:

$$\ln w_{ij/l} \overset{\text{approx.}}{\sim} \mathcal{N}\left(\mu_{w_{ij/l}}, (\sigma_{w_{ij/l}})^2\right),$$

where:

$$\mu_{w_{ij/l}} = \ln M_{1/l} - \frac{1}{2}\ln(1 + \frac{M_{2/l}}{M_{1/l}^2}) \tag{30}$$

$$\sigma_{w_{ij/l}}^2 = \ln(1 + \frac{M_{2/l}}{M_{1/l}^2}) \tag{31}$$

$$M_{1/l} = \sum_{k \neq l}^{N} \exp\left(\mu_i^k + \mu_j^k + \frac{1}{2}\left[(\sigma_i^k)^2 + (\sigma_j^k)^2 + 2\sigma_{ij}^k\right]\right)$$

$$M_{2/l} = \sum_{k=1}^{N} \left[\exp\left((\sigma_i^k)^2 + (\sigma_j^k)^2 + 2\sigma_{ij}^k\right) - 1\right] \times$$
$$\exp\left(2(\mu_i^k + \mu_j^k) + (\sigma_i^k)^2 + (\sigma_j^k)^2 + 2\sigma_{ij}^k\right)$$

$\square$

**Proposition 2.1.** *For a pair of synaptic connections* $\mathbf{w} = (w_{ij}, w_{mn})^T$ *following a 2-dimensional log-normal distribution, the covariance between* $\ln w_{ij}$ *and* $\ln w_{mn}$ *is:*

$$\sigma_{w_{ij,mn}} = \ln(1 +$$
$$\frac{\sum_{k=1}^{N}\exp\left(\sum_{\phi}^{\{i,j,m,n\}}\mu_\phi^k + \frac{1}{2}\sum_{\phi}^{\{i,j,m,n\}}(\sigma_\phi^k)^2 + \sum_{\phi}^{\{ij,mn\}}\sigma_\phi^k\right)\left[\exp\left(\sum_{\phi}^{\{i,j\}\times\{m,n\}}\sigma_\phi^k\right)-1\right]}{\sum_{k,l=1}^{N}\exp\left(\sum_{\phi}^{\{i,j\}}\mu_\phi^k + \sum_{\phi}^{\{m,n\}}\mu_\phi^l + \frac{1}{2}\sum_{\phi}^{\{i,j\}}(\sigma_\phi^k)^2 + \frac{1}{2}\sum_{\phi}^{\{m,n\}}(\sigma_\phi^l)^2 + \sigma_{ij}^k + \sigma_{mn}^l\right)}).$$

**Proof.** We have assumed that the ensemble $\mathbf{w}$ approximately follows 2-dimensional log-normal distribution, which states:

$$\ln\mathbf{w} = \begin{pmatrix}\ln w_{ij}\\\ln w_{mn}\end{pmatrix} \overset{\text{approx.}}{\sim} \mathcal{N}(\mu_{\mathbf{w}} = \begin{pmatrix}\mu_{w_{ij}}\\\mu_{w_{mn}}\end{pmatrix}, \Sigma_{\mathbf{w}} = \begin{pmatrix}\sigma_{w_{ij}}^2 & \sigma_{w_{ij,mn}}\\\sigma_{w_{mn,ij}} & \sigma_{w_{mn}}^2\end{pmatrix}).$$

Given the property of a multivariate log-normal random variable[4], the covariance between two components of random variable $\mathbf{w}$, $\text{Cov}[w_{ij}, w_{mn}]$, and the covariance between their logarithms, $\sigma_{w_{ij,mn}}$, are related by the following equation:

$$\text{Cov}[w_{ij}, w_{mn}] = \left[\exp(\sigma_{w_{ij,mn}}) - 1\right]\exp\left(\mu_{w_{ij}} + \mu_{w_{mn}} + \frac{1}{2}(\sigma_{w_{ij}}^2 + \sigma_{w_{mn}}^2)\right),$$

which is equivalent to:

$$\sigma_{w_{ij,mn}} = \ln\left(1 + \frac{\text{Cov}[w_{ij}, w_{mn}]}{\exp\left[\mu_{w_{ij}} + \mu_{w_{mn}} + \frac{1}{2}(\sigma_{w_{ij}}^2 + \sigma_{w_{mn}}^2)\right]}\right) \tag{32}$$

---

[4]For multivariate random variable $\ln X \sim \mathcal{N}(\mu, \Sigma)$, $\text{Var}[X]_{ij} = (e^{\Sigma_{ij}} - 1)e^{\mu_i + \mu_j + \frac{1}{2}(\Sigma_{ii} + \Sigma_{jj})}$

Given that the expressions for $\mu_{w_{ij}}, \mu_{w_{mn}}$ and $\sigma^2_{w_{ij}}, \sigma^2_{w_{mn}}$ have been derived in equation 28 and 29, the problem reduces to finding $\mathrm{Cov}[w_{ij}, w_{mn}]$. From its definition, we have:

$$\mathrm{Cov}[w_{ij}, w_{mn}] = \mathbb{E}[w_{ij}w_{mn}] - \mathbb{E}[w_{ij}]\mathbb{E}[w_{mn}]$$

$$= \mathbb{E}\left[(\sum_{k=1}^{N}\mathbf{x}_i^k\mathbf{x}_j^k)(\sum_{l=1}^{N}\mathbf{x}_m^l\mathbf{x}_n^l)\right] - \mathbb{E}\left[\sum_{k=1}^{N}\mathbf{x}_i^k\mathbf{x}_j^k\right]\mathbb{E}\left[\sum_{l=1}^{N}\mathbf{x}_m^l\mathbf{x}_n^l\right]$$

$$= \sum_{k,l=1}^{N}\mathbb{E}\left[\mathbf{x}_i^k\mathbf{x}_j^k\mathbf{x}_m^l\mathbf{x}_n^l\right] - \sum_{k=1}^{N}\mathbb{E}\left[\mathbf{x}_i^k\mathbf{x}_j^k\right]\sum_{l=1}^{N}\mathbb{E}\left[\mathbf{x}_i^l\mathbf{x}_j^l\right]$$

$$= \left(\sum_{k\neq l\ k,l=1}^{N}\mathbb{E}\left[\mathbf{x}_i^k\mathbf{x}_j^k\mathbf{x}_m^l\mathbf{x}_n^l\right] + \sum_{k=1}^{N}\mathbb{E}\left[\mathbf{x}_i^k\mathbf{x}_j^k\mathbf{x}_m^k\mathbf{x}_n^k\right]\right) - \sum_{k,l=1}^{N}\mathbb{E}\left[\mathbf{x}_i^k\mathbf{x}_j^k\right]\mathbb{E}\left[\mathbf{x}_i^l\mathbf{x}_j^l\right]$$

$$= \sum_{k\neq l\ k,l=1}^{N}\mathbb{E}\left[\mathbf{x}_i^k\mathbf{x}_j^k\right]\mathbb{E}\left[\mathbf{x}_m^l\mathbf{x}_n^l\right] + \sum_{k=1}^{N}\mathbb{E}\left[\mathbf{x}_i^k\mathbf{x}_j^k\mathbf{x}_m^k\mathbf{x}_n^k\right] - \sum_{k,l=1}^{N}\mathbb{E}\left[\mathbf{x}_i^k\mathbf{x}_j^k\right]\mathbb{E}\left[\mathbf{x}_i^l\mathbf{x}_j^l\right]$$

$$= \sum_{k=1}^{N}\left(\mathbb{E}\left[\mathbf{x}_i^k\mathbf{x}_j^k\mathbf{x}_m^k\mathbf{x}_n^k\right] - \mathbb{E}\left[\mathbf{x}_i^k\mathbf{x}_j^k\right]\mathbb{E}\left[\mathbf{x}_m^k\mathbf{x}_n^k\right]\right). \tag{33}$$

The second-to-last step uses the independence condition between data different patterns.

To derive the term $\mathbb{E}\left[\mathbf{x}_i^k\mathbf{x}_j^k\mathbf{x}_m^k\mathbf{x}_n^k\right]$ in the last line, recall that $\mathbf{x}_i^k, \mathbf{x}_j^k, \mathbf{x}_m^k, \mathbf{x}_n^k$ are four components of the same multivariate log-normal random vector $\mathbf{x}^k$. Thus, the product $\mathbf{x}_i^k\mathbf{x}_j^k\mathbf{x}_m^k\mathbf{x}_n^k$ is log-normally distributed. Therefore, by extending the same reasoning in equation 24, we get:

$$\ln(\mathbf{x}_i^k\mathbf{x}_j^k\mathbf{x}_m^k\mathbf{x}_n^k) \sim \mathcal{N}\left(\sum_{\phi}^{\{i,j,m,n\}}\mu_\phi^k, \sum_{\phi}^{\{i,j,m,n\}}(\sigma_\phi^k)^2 + \sum_{\phi}^{\{ij,mn\}}2\sigma_\phi^k + \sum_{\phi}^{\{i,j\}\times\{m,n\}}2\sigma_\phi^k\right),$$

where the different summations are defined as follows:

$$\sum_{\phi}^{\{i,j,m,n\}}\mu_\phi^k = \mu_i^k + \mu_j^k + \mu_m^k + \mu_n^k,$$

$$\sum_{\phi}^{\{i,j,m,n\}}(\sigma_\phi^k)^2 = (\sigma_i^k)^2 + (\sigma_j^k)^2 + (\sigma_m^k)^2 + (\sigma_n^k)^2,$$

$$\sum_{\phi}^{\{ij,mn\}}2\sigma_\phi^k = 2\sigma_{ij}^k + 2\sigma_{mn}^k, \quad \sum_{\phi}^{\{i,j\}\times\{m,n\}}2\sigma_\phi^k = 2\sigma_{im}^k + 2\sigma_{in}^k + 2\sigma_{jm}^k + 2\sigma_{jn}^k.$$

Consequently, the mean of this quadruple product term is:

$$\mathbb{E}\left[\mathbf{x}_i^k\mathbf{x}_j^k\mathbf{x}_m^k\mathbf{x}_n^k\right] = \exp\left(\sum_{\phi}^{\{i,j,m,n\}}\mu_\phi^k + \frac{1}{2}\sum_{\phi}^{\{i,j,m,n\}}(\sigma_\phi^k)^2 + \sum_{\phi}^{\{ij,mn\}}\sigma_\phi^k + \sum_{\phi}^{\{i,j\}\times\{m,n\}}\sigma_\phi^k\right). \tag{34}$$

For the term $\mathbb{E}\left[\mathbf{x}_i^k\mathbf{x}_j^k\right]\mathbb{E}\left[\mathbf{x}_m^k\mathbf{x}_n^k\right]$ in the equation 33, given that $\mathbb{E}\left[\mathbf{x}_i^k\mathbf{x}_j^k\right]$ and $\mathbb{E}\left[\mathbf{x}_m^k\mathbf{x}_n^k\right]$ were obtained as components in the summation in equation 26, we get:

$$\mathbb{E}\left[\mathbf{x}_i^k\mathbf{x}_j^k\right]\mathbb{E}\left[\mathbf{x}_m^k\mathbf{x}_n^k\right] = \exp\left(\sum_{\phi}^{\{i,j,m,n\}}\mu_\phi^k + \frac{1}{2}\sum_{\phi}^{\{i,j,m,n\}}(\sigma_\phi^k)^2 + \sum_{\phi}^{\{ij,mn\}}\sigma_\phi^k\right). \tag{35}$$

Substituting the results from equations 34 and 35 into equation 33, we obtain the covariance between two synaptic weights:

$$\text{Cov}[w_{ij}, w_{mn}] = \sum_{k=1}^{N} \exp\left(\sum_{\phi}^{\{i,j,m,n\}} \mu_{\phi}^{k} + \frac{1}{2}\sum_{\phi}^{\{i,j,m,n\}} (\sigma_{\phi}^{k})^2 + \sum_{\phi}^{\{ij,mn\}} \sigma_{\phi}^{k}\right) \times$$
$$\left[\exp\left(\sum_{\phi}^{\{i,j\}\times\{m,n\}} \sigma_{\phi}^{k}\right) - 1\right]. \tag{36}$$

Finally, by combining the covariance $\text{Cov}[w_{ij}, w_{mn}]$ with equations 28, 29, 32, we obtain an expression for $\sigma_{w_{ij,mn}}$ as a function of the parameters characterizing the data pattern distributions:

$$\sigma_{w_{ij,mn}} = \ln(1 +$$
$$\frac{\sum_{k=1}^{N} \exp\left(\sum_{\phi}^{\{i,j,m,n\}} \mu_{\phi}^{k} + \frac{1}{2}\sum_{\phi}^{\{i,j,m,n\}} (\sigma_{\phi}^{k})^2 + \sum_{\phi}^{\{ij,mn\}} \sigma_{\phi}^{k}\right) \left[\exp\left(\sum_{\phi}^{\{i,j\}\times\{m,n\}} \sigma_{\phi}^{k}\right) - 1\right]}{\sum_{k,l=1}^{N} \exp\left(\sum_{\phi}^{\{i,j\}} \mu_{\phi}^{k} + \sum_{\phi}^{\{m,n\}} \mu_{\phi}^{l} + \frac{1}{2}\sum_{\phi}^{\{i,j\}} (\sigma_{\phi}^{k})^2 + \frac{1}{2}\sum_{\phi}^{\{m,n\}} (\sigma_{\phi}^{l})^2 + \sigma_{ij}^{k} + \sigma_{mn}^{l}\right)}).$$
$$\tag{37}$$

$\square$

**Proposition 2.2.** *Under the approximation that both* $\mathbf{w} := (w_{ij}, w_{mn})^{T}$ *and* $\mathbf{w}_{/l} := (w_{ij/l}, w_{mn/l})^{T} = (\sum_{k\neq l}^{N} \mathbf{x}_i^k \mathbf{x}_j^k, \ \sum_{k\neq l}^{N} \mathbf{x}_m^k \mathbf{x}_n^k)^{T}$ *follow a multivariate log-normal distribution, i.e.* $\ln \mathbf{w} \sim \mathcal{N}(\mu_{\mathbf{w}}, \Sigma_{\mathbf{w}})$ *and* $\ln \mathbf{w}_{/l} \sim \mathcal{N}(\mu_{\mathbf{w}_{/l}}, \Sigma_{\mathbf{w}_{/l}})$, *the mutual information between the pair of connections* $\mathbf{w}$ *and a data pattern* $\mathbf{x}^l$ *can be expressed as:*

$$MI(\mathbf{w}; \mathbf{x}^l) = (\mu_{w_{ij}} + \mu_{w_{mn}} - \mu_{w_{ij/l}} - \mu_{w_{mn/l}}) + \frac{1}{2}\ln|\Sigma_{\mathbf{w}}| - \frac{1}{2}\ln|\Sigma_{\mathbf{w}_{/l}}|,$$

*where the expressions for* $\mu_{w_{ij}}$, $\mu_{w_{mn}}$ *are given in equation 28, and for* $\mu_{w_{ij/l}}$, $\mu_{w_{mn/l}}$ *in equation 30. For matrix* $\Sigma_{\mathbf{w}}$, *the diagonal elements are given by equation 29, and the off-diagonal elements by equation 37; for* $\Sigma_{\mathbf{w}_{/l}}$, *the corresponding expressions are provided by equations 31 and 38. Here,* $|\Sigma_{\mathbf{w}}|$ *denotes the determinant of the matrix* $\Sigma_{\mathbf{w}}$. *All information values are expressed in nats.*

**Proof.** Given that $\mathbf{w}$ follows a 2-dimensional log-normal variable, its differential entropy (in nats) can be written as[5]:

$$h(\mathbf{w}) = \frac{1}{2}\ln\left((2\pi e)^2 |\Sigma_{\mathbf{w}}|\right) + \mu_{w_{ij}} + \mu_{w_{mn}}.$$

For $h(\mathbf{w}|\mathbf{x}^l)$, observe that $p(\mathbf{w}|\mathbf{x}^l)$ is a shifted version of $p(\mathbf{w}_{/l}) := p(\sum_{k\neq l}^{N} \mathbf{x}_i^k \mathbf{x}_j^k, \ \sum_{k\neq l}^{N} \mathbf{x}_m^k \mathbf{x}_n^k)$. Therefore, the two distributions have the same entropy. This leads to:

$$h(\mathbf{w}|\mathbf{x}^l) = h(\mathbf{w}_{/l}) = \frac{1}{2}\ln\left((2\pi e)^2 |\Sigma_{\mathbf{w}_{/l}}|\right) + \mu_{w_{ij/l}} + \mu_{w_{mn/l}}.$$

The expressions for $\mu_{w_{ij/l}}, \mu_{w_{mn/l}}$, and for the diagonal elements of the covariance matrix, $\sigma^2_{w_{ij/l}}, \sigma^2_{w_{mn/l}}$, are given in the proof of Proposition 1.2. For the off-diagonal element $\sigma_{w_{ij,mn/l}}$, its derivation mirrors that of $\sigma_{w_{ij,mn}}$ in Proposition 2.1. For clarity, we present the explicit formula here:

$$\sigma_{w_{ij,mn/l}} = \ln(1 +$$
$$\frac{\sum_{k\neq l}^{N} \exp\left(\sum_{\phi}^{\{i,j,m,n\}} \mu_{\phi}^{k} + \frac{1}{2}\sum_{\phi}^{\{i,j,m,n\}} (\sigma_{\phi}^{k})^2 + \sum_{\phi}^{\{ij,mn\}} \sigma_{\phi}^{k}\right) \left[\exp\left(\sum_{\phi}^{\{i,j\}\times\{m,n\}} \sigma_{\phi}^{k}\right) - 1\right]}{\sum_{k,g\neq l}^{N} \exp\left(\sum_{\phi}^{\{i,j\}} \mu_{\phi}^{k} + \sum_{\phi}^{\{m,n\}} \mu_{\phi}^{g} + \frac{1}{2}\sum_{\phi}^{\{i,j\}} (\sigma_{\phi}^{k})^2 + \frac{1}{2}\sum_{\phi}^{\{m,n\}} (\sigma_{\phi}^{g})^2 + \sigma_{ij}^{k} + \sigma_{mn}^{g}\right)}).$$
$$\tag{38}$$

With $h(\mathbf{w})$ and $h(\mathbf{w}|\mathbf{x}^l)$ computed, we can now obtain the mutual information between any pair of synaptic interconnections and a data pattern:

$$MI(\mathbf{w}; \mathbf{x}^l) = h(\mathbf{w}) - h(\mathbf{w}|\mathbf{x}^l) = (\mu_{w_{ij}} + \mu_{w_{mn}} - \mu_{w_{ij/l}} - \mu_{w_{mn/l}}) + \frac{1}{2}\ln|\Sigma_{\mathbf{w}}| - \frac{1}{2}\ln|\Sigma_{\mathbf{w}_{/l}}|.$$

---

[5]For $n$-dimensional multivariate random variable $\ln X \sim \mathcal{N}(\mu, \Sigma)$, its differential entropy $h(X) = \frac{1}{2}\ln((2\pi e)^n |\Sigma|) + \sum_i \mu_i$ in nats.

$\square$

**Proposition 3.** *For a synaptic ensemble of $n$ connections, $\mathbf{w} := (w_{ij(1)}, w_{ij(2)}, ..., w_{ij(n)})^T$, where each $w_{ij(k)}$ denotes the connection between neuron $i(k)$ and $j(k)$, and under the approximation that both $\mathbf{w}$ and $\mathbf{w}_{/l} := (w_{ij(1)/l}, w_{ij(2)/l}, ..., w_{ij(n)/l})^T$ follow a multivariate log-normal distribution, i.e. $\ln \mathbf{w} \sim \mathcal{N}(\mu_{\mathbf{w}}, \Sigma_{\mathbf{w}})$ and $\ln \mathbf{w}_{/l} \sim \mathcal{N}(\mu_{\mathbf{w}_{/l}}, \Sigma_{\mathbf{w}_{/l}})$, the mutual information between the joint activity of these $n$ synaptic connections $\mathbf{w}$ and a data pattern $\mathbf{x}^l$ is given by:*

$$MI(\mathbf{w}; \mathbf{x}^l) = \sum_{k=1}^{n} \left( \mu_{w_{ij(k)}} - \mu_{w_{ij(k)/l}} \right) + \frac{1}{2} \ln |\Sigma_{\mathbf{w}}| - \frac{1}{2} \ln |\Sigma_{\mathbf{w}_{/l}}|.$$

*where the expressions for $\mu_{w_{ij(k)}}$ are given in equation 28, and for $\mu_{w_{ij(k)/l}}$ in equation 30. The diagonal and off-diagonal elements of $\Sigma_{\mathbf{w}}$ are given by equations 29 and 37, respectively; for $\Sigma_{\mathbf{w}_{/l}}$, by equations 31 and 38, respectively. All information values are expressed in nats.*

**Proof.** Based on the properties of an $n$-dimensional log-normal variable, the differential entropy of $\mathbf{w}$ is given by:

$$h(\mathbf{w}) = \frac{1}{2} \ln \left( (2\pi e)^n |\Sigma_{\mathbf{w}}| \right) + \sum_{k=1}^{n} \mu_{w_{ij(k)}}.$$

Meanwhile, $p(\mathbf{w}|\mathbf{x}^l)$ and $p(\mathbf{w}_{/l})$ have the same entropy owing to their shifting relation. Therefore, we have:

$$h(\mathbf{w}|\mathbf{x}^l) = h(\mathbf{w}_{/l}) = \frac{1}{2} \ln \left( (2\pi e)^n |\Sigma_{\mathbf{w}_{/l}}| \right) + \sum_{k=1}^{n} \mu_{w_{ij(k)/l}}.$$

Consequently, the information stored in ensemble $\mathbf{w}$ about pattern $\mathbf{x}^l$ is:

$$MI(\mathbf{w}; \mathbf{x}^l) = h(\mathbf{w}) - h(\mathbf{w}|\mathbf{x}^l) = \sum_{k=1}^{n} \left( \mu_{w_{ij(k)}} - \mu_{w_{ij(k)/l}} \right) + \frac{1}{2} \ln |\Sigma_{\mathbf{w}}| - \frac{1}{2} \ln |\Sigma_{\mathbf{w}_{/l}}|.$$

Since the derivation in Proposition 1.1, 1.2 is valid for any single connection $w_{ij(k)}$, and that in Proposition 2.1 for any pair, the expressions derived earlier can be directly applied to the quantities in this equation. $\square$

# B  Appendix: Simulation setup

After obtaining the analytical solution for the information encoded in synaptic connections, we performed a series of exploratory analyses by adjusting the parameter set $\mathcal{S}$ to examine its effect on $MI$.

We conducted three sets of simulations. The first set systematically varies parameters under constrained cases to reveal continuous dependencies. The second and third sets relax these constraints, exploring more general configurations by randomly sampling parameter combinations and employing regression analysis on the resulting data. Below, we describe the key differences in the constraints used for each simulation set to facilitate understanding of the subsequent results. For further details regarding the simulation setups, see the Methods section.

**First Simulation Set:** For different data patterns, the covariance matrix $\Sigma^k$ was held constant and constrained to be exchangeable: all diagonal elements were set to $\tilde{\sigma}^2$, and off-diagonal elements to $\tilde{\rho}\tilde{\sigma}^2$. Results of this analysis are shown in Figures 2A, 2B, 3A, and 3B.

**Second Simulation Set:** Again, $\Sigma^k$ was fixed across patterns. To explore more general covariance structures, an additional component proportional to $\mathbf{A}^T\mathbf{A}$, where $\mathbf{A}$ is a stochastic matrix, was added to the exchangeable component. Results are shown in Figures 2C and 3C.

**Third Simulation Set:** In this case, the covariance matrix $\Sigma^k$ varied between patterns. To avoid biases in matrix generation, each $\Sigma^k$ was constructed by generating random eigenvalues and random orthogonal matrices. Results of the corresponding analyses are shown in Figures 2D, 3D, and 4.

We also note that, while mutual information in the theoretical section is measured in nats for mathematical elegance, in the following analysis sections it is reported in bits, to provide a more intuitive understanding of the information quantity.

## B.1  Parameter setup for simulation

In this section, we provide a detailed explanation of the three sets of simulations used in our analyses:

**First Simulation Set:** We generated $N$ independent data patterns, $\{\mathbf{x}^1, \ldots, \mathbf{x}^N\}$, where each data pattern $\mathbf{x}^k$ was a $d$-dimensional random variable following a multivariate log-normal distribution: $\ln(\mathbf{x}^k) \sim \mathcal{N}(\mu^k, \Sigma^k)$. The components of each $\mu^k$ were sampled from a uniform distribution between -1 and 1, and the covariance matrix was modeled as an exchangeable matrix:

$$\Sigma^1 = \Sigma^2 = ... = \Sigma^N = \begin{pmatrix} \tilde{\sigma}^2 & \tilde{\rho}\tilde{\sigma}^2 & \cdots & \tilde{\rho}\tilde{\sigma}^2 \\ \tilde{\rho}\tilde{\sigma}^2 & \tilde{\sigma}^2 & \cdots & \tilde{\rho}\tilde{\sigma}^2 \\ \vdots & \vdots & \ddots & \vdots \\ \tilde{\rho}\tilde{\sigma}^2 & \tilde{\rho}\tilde{\sigma}^2 & \cdots & \tilde{\sigma}^2 \end{pmatrix}.$$

The dimension $d$ was chosen to be 20.

We varied the parameters $N$, $n$, $\tilde{\sigma}$, and $\tilde{\rho}$ across different analyses. In Figure 2A, we set $n = 1$, $\tilde{\rho} = 0$, systematically varied $N$ and $\tilde{\sigma}$, and calculated the corresponding values of $MI(\mathbf{x}^1; w_{23})$. In Figure 2B, we set $n = 1$, $\tilde{\sigma} = 1$, varied $N$ and $\tilde{\rho}$, and again calculated $MI(\mathbf{x}^1; w_{23})$. In Figure 3A, we set $N = 10$, and $\tilde{\rho} = 0$, varied the size $n$ by incrementally adding randomly selected connections to the synaptic ensemble $\mathbf{w}$, as well as varied $\tilde{\sigma}$, and computed the corresponding values of $MI(\mathbf{x}^1; \mathbf{w})$. In Figure 3B, we set $N = 10$, and $\tilde{\sigma} = 1$, varied $n$ in the same way as above, as well as varied $\tilde{\rho}$, and computed $MI(\mathbf{x}^1; \mathbf{w})$. In all cases where $\tilde{\rho}$ was varied, it was restricted to the range $\frac{-1}{d-1} \leq \tilde{\rho} \leq 1$. This constraint arises from the requirement that the covariance matrices remain positive semi-definite. Specifically, the eigenvalues of the exchangeable covariance matrix $\Sigma^k$ are $\tilde{\sigma}^2(1-\tilde{\rho})$ with multiplicity $d - 1$, and $\tilde{\sigma}^2(1 + (d - 1)\tilde{\rho})$ with multiplicity 1; ensuring non-negativity for all eigenvalues imposes the stated range on $\tilde{\rho}$.

**Second Simulation Set:** The objective of this simulation set was to sample multiple distribution-information "tuples", i.e., $\left(\{\mathbf{x}^1, \ldots, \mathbf{x}^N\}, \mathbf{w}, MI(\mathbf{w}; \mathbf{x}^l)\right)$, and to analyze potential relationships between $MI$ and distribution parameters based on these samples. This allowed for a more general exploration of the dependence of $MI$ on the parameters than the one with the first simulation set, which only varied the values of specific individual parameters. To this end, in this set, instead of systematically varying parameters across ranges, all distribution parameters were sampled randomly.

The dimension was set to $d = 20$, and the number of patterns $N$ was randomly selected from 2 to 50 with equal probability. The parameters $\{\mu^1, \ldots, \mu^N\}$ were sampled from a uniform distribution between -1 and 1 for each component. The covariance matrices were identical for all patterns but more general than the previous set:

$$\Sigma^1 = \Sigma^2 = ... = \Sigma^N = \mathbf{A}^T\mathbf{A}/d + \begin{pmatrix} \tilde{\sigma}^2 & \tilde{\rho}\tilde{\sigma}^2 & \ldots & \tilde{\rho}\tilde{\sigma}^2 \\ \tilde{\rho}\tilde{\sigma}^2 & \tilde{\sigma}^2 & \ldots & \tilde{\rho}\tilde{\sigma}^2 \\ \vdots & \vdots & \ddots & \vdots \\ \tilde{\rho}\tilde{\sigma}^2 & \tilde{\rho}\tilde{\sigma}^2 & \ldots & \tilde{\sigma}^2 \end{pmatrix}.$$

Here, $\tilde{\sigma}$ was drawn from the absolute values of a standard normal distribution, and $\tilde{\rho}$ was sampled from a uniform distribution in the range $[\frac{-1}{d-1}, 1]$. The matrix $\mathbf{A}$ is stochastic, with each element sampled from a standard normal distribution. Both components, $\mathbf{A}^T\mathbf{A}/d$ and the exchangeable covariance matrix, are positive semi-definite, ensuring that their sum is a valid covariance matrix. The introduction of $\mathbf{A}^T\mathbf{A}/d$ adds degrees of freedom, but it tends to produce matrices with larger diagonal elements, potentially neglecting distributions with highly correlated components. In contrast, the original exchangeable matrix can capture high correlations between components due to its structure, but it lacks sufficient degrees of freedom to account for more complex variability in the data. The $1/d$ scaling factor was designed to prevent either component from dominating. Consequently, by combining the two components, the resulting covariance matrix strikes a balance between flexibly capturing a wider range of data patterns, and the ability to model correlations, thereby providing a richer and more balanced representation of the underlying data distributions.

Using this setup, we generated 1000 samples of $\left(\{\mathbf{x}^1, \ldots, \mathbf{x}^N\}, \mathbf{w}, MI(\mathbf{w}; \mathbf{x}^l)\right)$. In cases in which the computed $MI(\mathbf{w}; \mathbf{x}^l)$ values were invalid - i.e., NaN or infinity, an extrapolation method was used to estimate $MI$ (see Section B.2). For Figure 2C, we analyzed the regression between $MI$ and $N$. For Figure 3C, we analyzed the regression between $MI$ and $n$.

**Third Simulation Set:** The third set of simulations followed a similar approach to the second in that we sampled numerous distribution-information "tuples" $\left(\{\mathbf{x}^1, \ldots, \mathbf{x}^N\}, \mathbf{w}, MI(\mathbf{w}; \mathbf{x}^l)\right)$, the dimension $d$ was set to 20, the number of patterns $N$ was randomly selected from 2 to 50 with equal probability, and the parameters $\{\mu^1, \ldots, \mu^N\}$ were sampled from a uniform distribution between -1 and 1 for each component. The key difference was that the covariance matrices varied across patterns, following:

$$\Sigma^k = \mathbf{Q}\mathbf{\Lambda}\mathbf{Q}^T + r\mathbf{1}\mathbf{1}^T.$$

Here, $\mathbf{Q}$ is a random orthogonal matrix generated using the common practice of performing QR decomposition on a matrix whose elements were drawn from a standard normal distribution. $\mathbf{\Lambda}$ is a diagonal matrix with random positive eigenvalues sampled from a normal distribution (mean 0.01, standard deviation 1.2), which is slightly broader than the standard normal to enable richer representations. The scalar factor $r$ was sampled from a uniform distribution in the range [0, 1.44] to prevent either component from dominating. Similar to the second simulation set, by combining these two components in the construction of the covariance matrices, we captured both the flexibility offered by $\mathbf{Q}\mathbf{\Lambda}\mathbf{Q}^T$ and the structure in $\mathbf{1}\mathbf{1}^T$ needed to model strong correlations across components.

Using this setup, we generated 1000 samples. In Figure 2D, we analyzed the regression between $MI$ and $N$, while in Figure 3D, we analyzed the regression between $MI$ and $n$. For the analysis of $MI/n$ in Figure 4, we generated an additional 20,000 samples. For all cases, we applied the extrapolation method to handle any invalid $MI$ values.

## B.2   Extrapolation method for simulation

During the generation of the data tuples $\left(\{\mathbf{x}^1, \ldots, \mathbf{x}^N\}, \mathbf{w}, MI(\mathbf{w}; \mathbf{x}^l)\right)$ described above, the $MI$ values obtained can, at times, be NaN or infinity due to errors from the approximations made in our propositions (for instance, that the sum of log-normal variables are also distributed log-normally). In such instances of invalid $MI$ values, we applied the following extrapolation technique to estimate a valid value of $MI$. In the datasets used for our analyses, only $\approx 1\%$ of the samples were obtained using this extrapolation method.

To measure similarity between samples $\left(\{\mathbf{x}^1, \ldots, \mathbf{x}^N\}, \mathbf{w}\right)$, we extracted six key features that capture the most prominent characteristics of each sample, and used them to compute Euclidean distance between samples (in 6-D space). The six features are:

1. Number of data patterns ($N$);
2. Size of the synaptic ensemble $\mathbf{w}$ ($n$);
3. Pattern centroid dispersion ($q_1 = \sum_k (\mu^k - \frac{1}{N} \sum_i \mu^i)^2$), which reflects the spread among the centroids of different patterns;
4. Average determinant of covariance matrices ($q_2 = \frac{1}{N} \sum_k |\Sigma^k|$), representing the overall breadth within each pattern distribution;
5. Average ratio of the largest to smallest eigenvalues ($q_3 = \frac{1}{N} \sum_k (\lambda_{\max}^k / \lambda_{\min}^k)$), which reflects the degree of data "stretching" or anisotropy;
6. Correlation ratio ($q_4 = \frac{1}{N} \frac{\sum_{i \neq j} |\Sigma_{ij}^k|}{\sum_{i=j} |\Sigma_{ij}^k|}$), representing the average ratio between the sum of off-diagonal elements and diagonal elements, which indicates the correlation structure among the pattern distributions.

We normalized these six features — $(N, n, q_1, q_2, q_3, q_4)$ — to account for differences in their inherent scales, thereby ensuring that each feature contributes equally to the distance calculation. The most similar sample to the current target sample, defined as the one with the smallest distance according to our metric, was identified, and its $MI$ value was used as the extrapolated substitute.

