# OpenReview forum: "Quantifying information stored in synaptic connections rather than in firing activities of neural networks"
_NeurIPS.cc/2025/Workshop/UniReps — UniReps2025_

### Official Review · Reviewer_98yU · 2025-09-07

**Confidence:** 3

**Review:**

### Summary
This work analytically derives the mutual information between data that follow log-normal distributions and synaptic connections in a continuous Hopfield network, which is learned through the Hebbian rule. Further, the numerical experiments demonstrate that (i) as the number of stored patterns increases, the information carried by each synapse decreases, supporting the distributed coding hypothesis; (ii) as the number of synapses increases, the mutual information increases; (iii) information synergy, where the synapses together encode more information than the sum of the individuals.

### Strength
* This is a well-written work that addresses an important gap in quantifying information stored by a network's synapses. The existing literature, as the authors point out, assumes independence of synapses or quantifying individual synaptic information by averaging.

### Weakness
* Section 3.4 addresses the corresponding weakness sufficiently, such as the log-normal assumption for data being restrictive and the work only looks at a continuous Hopfield network. However, both are fair assumptions and the scope of the work is well under that of an extended abstract.

### Minor suggestion & questions
* Section 3.4 can be reframed as Discussion, but this is very minor
* In line 133 and relevant figures, why $x^1$ specifically? Is it just an arbitrary example?
* Section 3.3 seems to be the most interesting result to me and can be highlighted upfront, and 3.1+3.2 can be reasonably combined in a future edition of this work.

**Score:**

4

**Topic Fit:**

2

---

### Official Review · Reviewer_amU8 · 2025-09-10
**The idea of the paper is interesting. The results are also interesting. Further analytical treatment and computational experiments could strengthen the paper.**

**Confidence:** 3

**Review:**

The paper develops an analytical information theoretic framework about how much information is stored in synaptic weghts instead of firing activities of neurons using a continous Hopfield network with Hebbian weights. The authors model data patterns as independent multi-variate log-normal variables approximate sums of products via the Fenton-Wilkinson log-normal approximation, and derived closed form expressions for mutual information between arbitrary synaptic ensembles (singletones, pairs, and n-tuples) and a chosen pattern. Simulation studies explores how mutual information scales with number of stored patterns, ensemble size, and covarinace parameters. They report: i) the existence of distributed coding meaning that per synaptic information decreases with number of stored patterns, ii) total mutual information grows superlinearly in ensemble size leading to a weight based capacity scaling, iii) synergy among synapses: the joint mutual information exceeds the sum of single-synapse mutual informations. I believe its overall strengths are: 1) investigating a novel and interesting question, 2) analytical derivations for ensembles, 3) interesting empirical findings, 4) clear link to prior works and good framing. Moreover, its main weaknesses include: 1) heavy reliance on approximation such as sum of log-normals is eqaul to log-normal, 2) strong modelling assumption about data distributions and independence, 3) limited empirical validation of analytical mutual information formula. I believe in general for an extended abstract this a well-crafted paper and should be accepted.

**Score:**

4

**Topic Fit:**

2

---

### Official Review · Reviewer_u8zm · 2025-09-13
**A Novel Information-Theoretic Framework for Quantifying Information Storage in Synaptic Weights**

**Confidence:** 3

**Review:**

**Summary:**
This paper introduces a novel theoretical framework to quantify the information stored in the synaptic weights of a neural network, shifting the focus from the traditional analysis of neural activations. The authors study a simplified setting of a continuous Hopfield network with Hebbian learning and model the memories to be stored as multivariate log-normal distributions. This allows them to derive a closed-form analytical expression for the mutual information between the input patterns and the approximated distribution of synaptic weights resulting from Hebbian learning. The authors leverage this formula to lay the foundation for an information-theoretic analysis of synaptic coding, providing quantitative insights into several core concepts. Specifically, they: (1) validate the idea of distributed coding by demonstrating that information per synapse decreases as memory load increases; (2) use the framework to analyze the network's storage capacity; and (3) most significantly, demonstrate the existence of information synergy, where the average information contributed by a synapse increases with the size of the ensemble being considered.

**Strengths:**
- High Originality and Significance: The paper provides a tractable mathematical description for an important and often underexplored question: how to formally measure the information encoded in a network’s parameters, rather than its activations. This opens up new and valuable avenues for studying learning and memory storage in both biological and artificial networks.

- Technical Quality and Clarity: The derivation of a closed-form expression for the mutual information between input patterns and synaptic weight ensembles is a significant technical achievement. This analytical tractability allows the authors to generate clean, interpretable results (e.g., Figures 2A, 3A) and to investigate the effects of data statistics on the resulting synapses from an information-theoretic perspective. The paper is clearly written and logically structured.

- Important Insights: The framework successfully formalizes established intuitions while also uncovering non-obvious results. The quantitative demonstration of distributed coding is valuable, but the finding of information synergy is particularly insightful. The result that the average "information per synapse" increases with ensemble size (Figure 4) provides strong evidence that synaptic weights form a complex, cooperative code where the whole is greater than the sum of its parts.

**Weaknesses / Suggestions for Improvement:**

- Limited Scope of Model Assumptions: The framework's elegance and analytical tractability come at the cost of highly specific assumptions (continuous Hopfield network, Hebbian rule, log-normal distributions, and the Fenton-Wilkinson approximation). While this is an acceptable trade-off for a foundational study (and the assumptions are not without justification, as the authors note that log-normal-like distributions are common in biological systems) it does limit the immediate applicability to modern deep learning and more complex biological settings. The paper would be strengthened by a more thorough discussion of these limitations and, if possible, an exploration of the robustness of its qualitative findings (especially synergy). This could potentially be explored via numerical simulations, which are not bound by the requirements for mathematical tractability, to test different distributional assumptions or learning rules.

- Connection to Modern Deep Learning: While the results are framed within the paradigm of memory storage, the connection to modern deep learning is limited. I would encourage the authors to add a short discussion speculating on how the principles uncovered here might apply to deep learning, where weights typically parameterize a function for information transformation (rather than direct memory storage) and are trained via gradient-based optimization. For instance, could this framework help us understand the distributed nature of feature representations in hidden layers or the information-theoretic effects of network pruning?

- Minor Point on Figure 1: While the conceptual model is clear, Figure 1 is overly schematic and its purpose in the narrative could be stronger. I would suggest augmenting it to better connect the two panels and visually ground the reader in the paper's setup. For example, panel A could be labeled to indicate the `d`-dimensional nature of the input patterns. Panel B could then more explicitly illustrate how the patterns from panel A result in a synaptic ensemble (the red connections) on the right via Hebbian update. This could be achieved by highlighting the specific neurons involved in a weight and perhaps schematically representing the resulting weight distribution, making the figure a more effective visual summary.

**Conclusion:**
Overall, this is a high-quality, original, and significant contribution. It introduces an elegant framework to study a fundamental question regarding the role of synaptic weights in memory storage, and the results, particularly on synergy, are non-trivial and insightful. Despite the simplified model, the work serves as a strong proof-of-concept and is highly likely to inspire future research.

**Score:**

4

**Topic Fit:**

2

---

### Official Review · Reviewer_WBbY · 2025-09-15
**Quantifying information stored in synaptic connections  rather than in firing activities of neural networks**

**Confidence:** 5

**Review:**

Summary:

The paper introduces a theoretical framework for quantifying information stored in synaptic connections of neural networks, as opposed to the well-studied firing activity of neurons. Using continuous Hopfield networks and modeling data patterns as independent multivariate log-normal distributions, the authors derive analytical expressions for the mutual information (MI) between synaptic connections and data patterns. They found that the information per synapse decreases as more patterns are stored, aligning with the principle of distributed coding. In the capacity analysis, the total stored information scales are leading to superlinear storage capacity compared to classical Hopfield models. The ensembles of synapses encode more information collectively than the sum of their parts, demonstrating information synergy at the synaptic level. The log-normal assumption is argued to generalize to many natural data modalities (e.g., vision, sound, language).

Pros:

++ The continuous Hopfield networks and Hebbian learning have longstanding ties to models of memory and cortical dynamics. And the distributed coding and synergy findings resonate with principles observed in brain function.

++ The usage of derivations, log-normal approximations (Fenton–Wilkinson method) make the problem mathematically approachable, and the explicit formulas for single synapses, pairs, and n-tuples strengthen generalizability.

++ The framework for quantifying information in synaptic weights is relatively underexplored as compared to firing activity, and the paper tries to move beyond heuristic notions of “information per synapse.”

Cons:

-- The choice of log-normal distribution for patterns, while justified, may oversimplify real-world data and neural weight statistics, and the robustness to multimodal or non-log-normal data remains speculative.

-- The framework quantifies potential information stored in weights, but does not address retrieval dynamics, which are significant for memory utility.

-- Hebbian learning is assumed, excluding modern plasticity rules or backpropagation. The paper by Schiller and colleagues, The Many Worlds of Plasticity Rules, may add more context

**Score:**

4

**Topic Fit:**

2